# On the Power of Decision Trees
# in Auto-Regressive Language Modeling

**Yulu Gan**
Massachusetts Institute of Technology
yulu@csail.mit.edu

**Tomer Galanti**
Texas A&M University
galanti@tamu.edu

**Tomaso Poggio**
Massachusetts Institute of Technology
tp@csail.mit.edu

**Eran Malach**
Harvard University
eran.malach@gmail.com

## Abstract

Originally proposed for handling time series data, Auto-regressive Decision Trees (ARDTs) have not yet been explored for language modeling. This paper explores both the theoretical and practical applications of ARDTs in this new context. We theoretically demonstrate that ARDTs can compute complex functions, such as simulating automata, Turing machines, and sparse circuits, by leveraging "chain-of-thought" computations. Our analysis provides bounds on the size, depth, and computational efficiency of ARDTs, highlighting their surprising computational power. Empirically, we train ARDTs on simple language generation tasks, showing that they can learn to generate coherent and grammatically correct text on par with a smaller Transformer model. Additionally, we show that ARDTs can be used on top of transformer representations to solve complex reasoning tasks. This research reveals the unique computational abilities of ARDTs, aiming to broaden the architectural diversity in language model development.

## 1  Introduction

In recent years, Large Language Models (LLMs) have achieved outstanding results in tasks such as natural language understanding, coding, and mathematical reasoning. LLMs predominantly utilize the Transformer architecture Vaswani et al. (2023), establishing it as the standard in this field. However, recent initiatives (Gu & Dao, 2023; Sun et al., 2023; Ma et al., 2023; De et al., 2024) have begun to challenge the dominance of Transformers. These alternatives, while not yet matching Transformer performance, offer advantages in terms of inference time efficiency. Moreover, some works are revisiting traditional non-neural network models for language modeling, such as classical symbolic models (Wong et al., 2023). These developments indicate a shift towards diverse, efficient, and interpretable language modeling methodologies.

Tree-based models, particularly favored for handling tabular data (Grinsztajn et al., 2022), continue to hold significant importance. While tree-based methods are mostly used for classification and regression tasks, Auto-regressive Decision Trees (ARDTs) (Meek et al., 2002) have been studied for time-series prediction, offering a simpler and more interpretable alternative to complex nonlinear approaches. Although the ARDT approach was not originally designed for language tasks, it has demonstrated considerable promise in various time-series datasets, outperforming traditional auto-regressive models while maintaining ease of interpretation. Motivated by these results, our study seeks to explore the potential of ARDTs for language prediction tasks, assessing whether they could serve as a viable, interpretable alternative to complex, resource-intensive language models.

Submitted to 38th Conference on Neural Information Processing Systems (NeurIPS 2024). Do not distribute.

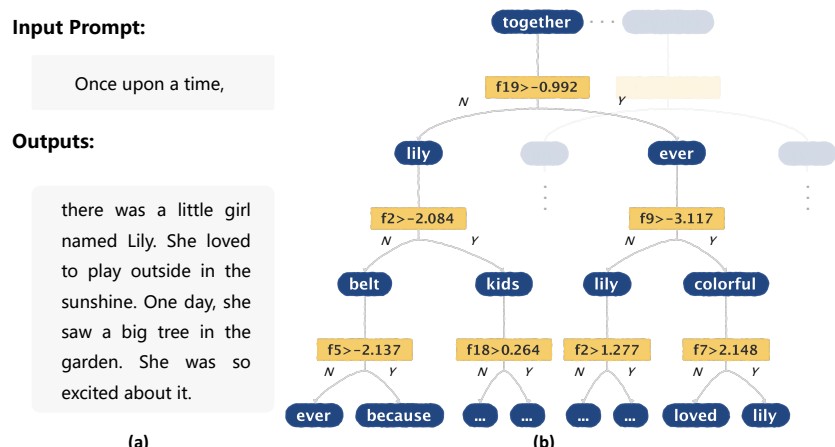

**Input Prompt:**

Once upon a time,

**Outputs:**

there was a little girl named Lily. She loved to play outside in the sunshine. One day, she saw a big tree in the garden. She was so excited about it.

(a)

(b)

Figure 1: **(a) An example of story continuation generated by our Auto-Regressive Decision Trees.** We use decision trees and, remarkably, attain results comparable to Transformer-based models in terms of linguistic fluency. **(b) The decision process of the decision trees.** We visualize part of the tree ensemble, and can observe which word is most relevant for the splitting rule at each node.

To understand the power of ARDTs, we first conduct theoretical studies demonstrating that ARDTs, using decision trees as next-token predictors, can compute more complex functions than traditional decision trees. We explore the classes of functions ARDTs can compute, showing their ability to simulate functions computed by automata, Turing machines, or sparse circuits through intermediate "chain-of-thought" computations. We provide bounds on the size, depth, and run-time (measured by the number of intermediate tokens) required for ARDTs to simulate these function classes. Our findings highlight the surprising computational capabilities of ARDTs, underscoring their potential as a powerful and interpretable alternative for language prediction tasks requiring complex function computations.

Our experimental results further demonstrate the practical utility of ARDTs in language generation tasks. Utilizing standard auto-regressive inference methods, these models generate output sequences token-by-token, appending each new token to the input of the subsequent iteration. When trained on the TinyStories dataset Eldan & Li (2023), ARDTs produce coherent and grammatically accurate text (see in Fig 1). Notably, decision tree ensembles with approximately 0.3 million parameters outperform a Transformer model with around 1 million parameters on the same Tinystories dataset, highlighting their efficiency despite a smaller size. We discuss our approach to training interpretable decision trees, which enhances the transparency of the decision-making process in language generation. Furthermore, we assess the ability of tree-based models to execute various logical reasoning tasks. Notably, tree ensembles built on top of transformer embeddings and trained on specific downstream tasks perform comparably to larger general models like InstructGPT Ouyang et al. (2022) and PaLM-540B Chowdhery et al. (2022), under the conditions of these particular tasks.

Our contribution can be summarized as follows:

- We extend the application of ARDTs to language prediction tasks, adopting a novel approach that capitalizes on their inherent simplicity and interpretability. This aims to broaden the architectural diversity in language model development.

- Through theoretical analysis, we demonstrate that ARDTs can compute a broader array of complex functions than previously recognized, including the simulation of automata, Turing machines, and sparse circuits. These theoretical findings deepen our understanding of ARDTs' computational capabilities.

- Our experimental results offer empirical evidence that ARDTs are capable of generating coherent and grammatically correct text, perform well compared to more complex models like small Transformers, and demonstrate solid reasoning abilities.

## 2 Related Work

**Decision Trees.** Tree based models have been widely used for solving different classification and regression tasks in machine learning (Navada et al., 2011). The ID3 algorithm was introduced by Quinlan (1986), and has been widely used for decision tree learning, along with the CART (Breiman et al., 1984; Lewis, 2000) algorithm. Decision tree ensembles, such as random forests (Breiman, 2001) and gradient boosted trees (Friedman, 2002), are also very popular. Despite continuous advancements in deep learning, decision tree ensembles still outperform neural network based models on tabular datasets (Shwartz-Ziv & Armon, 2022). Different from traditional decision trees, we use auto-regressive decision trees to perform language prediction tasks more efficiently.

**Learning Theory for Decision Trees.** There are a few theoretical works studying the power of decision trees in solving machine learning problems. The work of Brutzkus et al. (2020) shows that the ID3 algorithm can learn sparse functions in some setting. Kearns & Mansour (1996) show that decision trees are equivalent to boosting methods for amplifying the performance of weak learners on the distribution. Other works focus on other aspects of decision tree learnability (Rivest, 1987; Blum, 1992; Ehrenfeucht & Haussler, 1989; Bshouty & Burroughs, 2003). We note that from the approximation point of view, decision trees can be regarded as splines with free knots. For instance, piecewise constant hierarchical splines functions, similar to neural networks with threshold activation can also be seen as decision trees. Note that ReLU networks can be viewed as piecewise hierarchical linear splines (Anselmi et al., 2015; Yarotsky, 2016), and so decision trees can represent ReLU networks (see Aytekin (2022)), though possibly with an exponential number of parameters. We note that none of the works mentioned above studies the theory of auto-regressive decision trees, which is a novel contribution of our paper.

**Decision Trees for Language.** Despite gaining popularity in several fields of machine learning, tree based models are not widely used for language generation. Past works have utilized auto-regressive decision trees for time-series analysis (Meek et al., 2002), or use trees for basic language modeling (Potamianos & Jelinek, 1998). Decision trees were also used in parsing (Magerman, 1995; Heeman, 1999; Nallapati & Allan, 2002), modeling syntax (Filimonov, 2011) and language identification (Hakkinen & Tian, 2001).

## 3 Theory

To explore the capabilities of ARDTs, we initially undertake theoretical studies demonstrating that using decision trees as next-token predictors enables ARDTs to process significantly more complex functions than "standard" decision trees. Firstly, we define the theoretical setting of our analysis in Section 3.1. We then examine the various classes of functions that an ARDT can compute, as discussed in Sections 3.2, 3.3, and 3.4. Here, the computation involves the ARDT receiving an input sequence, such as a question, generating a series of intermediate tokens that describe the thought process, and finally producing the output token. Specifically, we demonstrate that functions computed by Automata, Turing machines, or sparse circuits can be emulated by an ARDT using these intermediate "chain-of-thought" computations. Additionally, we provide bounds on the size, depth, and runtime (measured by the number of intermediate tokens) required for ARDTs to simulate these classes of interest. Our findings affirm that ARDTs, by leveraging decision trees for next-token prediction, can handle far more complex functions than "standard" decision trees.

**Comment 1.** *The results in this section are representation results. That is, we study which functions can, in theory, be represented by auto-regressive decision trees. We do not provide any formal results on whether such functions can be learned from data. The question of how decision trees can be trained to produce "chain-of-thought" responses to input questions is beyond the scope of this work.*

### 3.1 Setting

We adapt the standard definition of a decision tree, as described by Quinlan (1986), to include modifications that allow for the processing of vector sequences of arbitrary lengths. Firstly, we establish a vocabulary $\mathbb{D}$, which serves as our token dictionary. Next, we define an input embedding $\Psi : \mathbb{D} \to \mathbb{R}^d$. For any sequence of tokens $\boldsymbol{s} \in \mathbb{D}^n$, $\Psi(\boldsymbol{s}) \in \mathbb{R}^{n \times d}$ represents the embedding applied individually to each token. The space comprising sequences of d-dimensional vectors is denoted by

$\mathcal{X} = \mathbb{R}^{* \times d}$. Subsequently, we define a decision tree $\mathcal{T}$ that receives an input $\boldsymbol{x} \in \mathcal{X}$ and outputs a token $y \in \mathbb{D}$.

In our experiments, detailed in Section 4, we apply a weighted-average operator to the word vectors of the sequence, where the average vectors are used as an input to the decision trees. For the theoretical analysis we study a different approach for using decision trees over vector sequences, where instead of averaging word vectors we "concatenate" them. That is, the decision tree is applied to the $L$ most recent words, in a "sliding window" fashion. We note that experimentally we observed that both the "sliding-window" and the weighted-average approach produced similar results, and use the weighted-average technique in our experiments for computational reasons.

We start by defining a decision tree $\mathcal{T}$ that gets inputs of a fixed length $L$, namely $\mathcal{T} : \mathbb{R}^{L \times d} \to \mathbb{D}$. We refer to the value $L$ as the *context length* of $\mathcal{T}$, and this value will correspond to the maximal length of a sequence that affects the computation of the tree. In this case, we treat the input $\boldsymbol{x} \in \mathbb{R}^{L \times d}$ as a vector, and let $\mathcal{T}$ be a standard decision tree operating on vectors of size $L \cdot d$. Namely, $\mathcal{T}$ is defined by a binary tree, where each node corresponds to an input feature $x_{i,j}$ and some threshold $\tau \in \mathbb{R}$. Each leaf corresponds to some output token $y \in \mathbb{D}$. The output of the tree $\mathcal{T}$ is computed by starting at the root, and for each internal node with feature $x_{i,j}$ and threshold $\tau$, moving to the right node if $x_{i,j} \geq \tau$ and otherwise moving to the left node. When reaching a leaf, we output the value $y \in \mathbb{D}$ corresponding to the leaf. The *size* of the tree $\mathcal{T}$ is the number of leaves in the tree, and its *depth* is the maximum length of a path from root to leaf. Note that the runtime of computing the output of $\mathcal{T}$ corresponds to the *depth* of the tree.

Now, given some tree over length-$L$ inputs $\mathcal{T} : \mathbb{R}^{L \times d} \to \mathbb{D}$, we apply $\mathcal{T}$ to an input of arbitrary length $\boldsymbol{x} \in \mathcal{X}$ using the following simple rule: if $\boldsymbol{x}$ has length shorter than $L$, we pad it to length $L$ by prepending the input, adding additional padding ($\langle \text{PAD} \rangle$) tokens at the beginning; if $\boldsymbol{x}$ is longer than $L$, we apply $\mathcal{T}$ only to the last $L$ tokens in $\boldsymbol{x}$. This induces a decision tree with arbitrary length inputs $\mathcal{T} : \mathcal{X} \to \mathbb{D}$.

Finally, we use the tree $\mathcal{T}$ as a next-token predictor function, applied over some input using auto-regressive computation. That is, we define a sequence-to-sequence predictor $\mathcal{T}^{\text{AR}} : \mathbb{D}^* \to \mathbb{D}^*$ induced from the tree $\mathcal{T}$ as follows: for every input $\boldsymbol{s} \in \mathbb{D}^n$, recursively define $s_{n+i+1} = \mathcal{T}(\Psi(s_1, \ldots, s_{n+i}))$, and let $\mathcal{T}^{\text{AR}}(s_1, \ldots, s_n) = (s_{n+1}, s_{n+2}, \ldots)$. We call $\mathcal{T}^{\text{AR}}$ an *auto-regressive decision tree* (ARDT).

In the rest of this section, we will analyze the capacity of ARDTs to simulate some function classes. Following Malach (2023), we give the following definition:

> **Definition 2.** *For some class $\mathcal{F}$ of functions $f : \mathbb{D}^n \to \mathbb{D}$, we say $\mathcal{F}$ can be simulated by auto-regressive decision-trees in length complexity $T$, if for every $f \in \mathcal{F}$ there exists $\mathcal{T}^{\text{AR}}$ s.t. for all $\boldsymbol{s} \in \mathbb{D}^n$, we have $\mathcal{T}_T^{\text{AR}}(\boldsymbol{s}) = f(\boldsymbol{s})$ (where $\mathcal{T}_T^{\text{AR}}$ indicates the output of $\mathcal{T}^{\text{AR}}$ at iteration $T$).*

In other words, we say that the tree $\mathcal{T}^{\text{AR}}$ can compute the function $f$, if given some input sequence $\boldsymbol{s}$, it generates $T$ tokens followed by the correct output $f(\boldsymbol{s})$. That is, we allow the tree to use $T$ intermediate tokens as "chain-of-thought" before outputting the correct answer.

## 3.2 Simulating Automata

An automaton $\mathcal{A}$ is defined over an alphabet $\Sigma$, using a set of states $Q$, an initial state $q_0 \in Q$ and a transition function $\delta : Q \times \Sigma \to Q$. We always assume that $|\Sigma| \geq 2$ and $|Q| \geq 2$. The automaton $\mathcal{A}$ gets an input string $\boldsymbol{x} \in \Sigma^*$, and computes an output state $\mathcal{A}(\boldsymbol{x}) \in Q$ by starting at state $q_0$ and at each iteration $i$ transitioning to the next state based on the $i$-th token $x_i$, namely $q_i = \delta(q_{i-1}, x_i)$. The automaton then returns the state reached at the final iteration.

Let $\mathcal{F}_n^{\text{Aut}}$ is the class of all functions computed by automata over strings of length $n$. Namely, $\mathcal{F}_n^{\text{Aut}}$ is the class of functions $f : \Sigma^n \to Q$ s.t. for all $f \in \mathcal{F}_n^{\text{Aut}}$ there exists an automaton $\mathcal{A}$ s.t. $\mathcal{A}(\boldsymbol{x}) = f(\boldsymbol{x})$ for all $\boldsymbol{x} \in \Sigma^n$.

The class of functions computed by Automata has been well-studied from the early days computer science theory (Hopcroft et al., 2001), and has various important connections to language problems. This class of functions is also interesting in the context of reasoning tasks for language modeling. For

example, the *Web-of-Lies* and *Navigate* problems in the Big-Bench Hard dataset (Srivastava et al., 2023) can be solved by finite state Automata.

We show that ARDTs can simulate Automata:

> **Theorem 3.** *Let $\mathbb{D} = \Sigma \cup Q \cup \{\langle \text{PAD} \rangle\}$. Then, $\mathcal{F}_n^{\text{Aut}}$ can be simulated by ARDTs of size $O\left(|\mathbb{D}|^2\right)$, depth $O(\log |\mathbb{D}|)$ and context length $L \geq n$, in length complexity $O(n)$.*

Note that ARDTs simulate Automata very efficiently: the total run-time of the ARDT guaranteed by Theorem 3 is $O(n \log |\mathbb{D}|)$, which corresponds to the time it takes to read all the input bits. In this sense, no algorithm can simulate Automata significantly faster than ARDT.

In the proof, we construct an ARDT that, at every iteration $i$, outputs the state of the Automaton at step $i$ (denoted $q_i$). The state at step $i + 1$ is only a function of the $i$-th state, given by the most recent token generated by the model; and the $i$-th input, which is always given by looking back $n + 1$ tokens. Therefore, a simple tree, applied as a *sliding-window* over the input, can compute the transition matrix to find the next state. The full proof is given in Appendix A.

Next, we show that the above result implies a *separation* between ARDTs and standard decision trees. Specifically, we show that if we use a decision-tree over the input to directly predict the final output of the Automata, without outputting intermediate states, then the size of the decision tree must be exponential in the length of the input:

> **Theorem 4.** *There exists some $f \in \mathcal{F}_n^{\text{Aut}}$ s.t. any decision tree that computes $f$ has size $\geq \Omega(2^n)$.*

This shows that the fact that ARDTs can perform intermediate computations auto-regressively (e.g., perform *chain-of-thought*) significantly improves their efficiency[1]. To prove the result, we show that computing the *parity* of a sequence of bits (i.e., whether the number of bits is even or odd) requires a tree of exponential size, but can be easily computed by a simple 2-state Automaton.

*Proof of Theorem 4.* Consider the binary alphabet $\Sigma = \{0, 1\}$ and the state set $Q = \{\text{even}, \text{odd}\}$, with $|\Sigma| = 2$ and $|Q| = 2$. We define a function $f : \Sigma^n \to Q$ as follows:

$$f(\mathbf{x}) = \begin{cases} \text{even} & \text{if } \sum x_i \mod 2 = 0, \\ \text{odd} & \text{otherwise.} \end{cases}$$

The function $f$ describes the parity of the sum of bits in $\mathbf{x}$ and can be efficiently computed by an automaton that toggles between states even and odd upon encountering a 1.

Suppose a decision tree $\mathcal{T}$ computes $f$. We claim that the size of $\mathcal{T}$ must be at least $2^n$. Assume for contradiction that $\mathcal{T}$ has fewer than $2^n$ leaves. Since $\mathcal{T}$ is a decision tree, we assume that all its leaves are reachable by some input $\mathbf{x} \in \{0, 1\}^n$.

Consider a leaf $l$ of $\mathcal{T}$ reached by some input $\mathbf{x}$, at a depth less than $n$. This implies that there exists at least one bit index $j \in [n]$ such that no decision node in $\mathcal{T}$ queries $x_j$ on the path to $l$. Define $\mathbf{x}' \in \{0, 1\}^n$ by flipping $x_j$ in $\mathbf{x}$, while keeping all other bits unchanged:

$$x_i' = \begin{cases} x_i & \text{if } i \neq j, \\ \neg x_j & \text{if } i = j. \end{cases}$$

Since $\mathbf{x}'$ alters $\mathbf{x}$ only at the unqueried index $j$, it follows the same path in $\mathcal{T}$ and reaches the same leaf $l$. Therefore, $\mathcal{T}(\mathbf{x}) = \mathcal{T}(\mathbf{x}')$. However, the definition of $f$ guarantees $f(\mathbf{x}) \neq f(\mathbf{x}')$ as their parities are different, leading to a contradiction. Thus $\mathcal{T}$ cannot compute $f$ with fewer than $2^n$ leaves. $\square$

---

[1]This is an example of how compositional sparsity can defeat the curse of dimensionality (Poggio, 2022). A function may not be approximated by a decision tree without an exponential number of parameters but may be represented efficiently by composing intermediate sparse functions, as ARDTs do.

### 3.3 Simulating Turing Machines

A Turing machine $\mathcal{M}$ is defined over an alphabet $\Sigma$, using a space set $Q$, initial state $q_0 \in Q$ and transition function $\delta : Q \times \Sigma \to Q \times \Sigma \times \{\langle \text{LEFT} \rangle, \langle \text{RIGHT} \rangle\}$. The Turing machine has a tape, where each cell contains a symbol from $\Sigma$. The *head* of the Turing machine is initialized at the leftmost cell on the tape in state $q_0 \in Q$. At each iteration of the machine, it reads the symbol $s \in \Sigma$ and given the head state $q \in Q$ uses $\delta(q, s)$ to determined the new state of the head, the symbol to write under the head, and whether to move the head left or right on the tape.

In our setting, we consider Turing machines with fixed memory $M$, i.e. Turing machines with access to a tape with $M$ cells. In particular, this means that the Turing machine $\mathcal{M}$ operate on inputs with $< M$ tokens. At the initial step, the input is written on the tape. If the input size is shorter than $M$, we add empty tokens $\{\emptyset\} \in \Sigma$ after the input sequence. We consider Turing machines with fixed runtime $T$, namely we let the machine run for $T$ iterations and then halt it. The output of the machine is the rightmost symbol on the tape after $T$ iterations. So, we define $\mathcal{M} : \Sigma^M \to \Sigma$ to be the function computed by the machine after $T$ steps. We denote by $\mathcal{F}_{M,T}^{\text{Turing}}$ the class of functions computed by Turing machines with memory of size $M$ and runtime $T$.

**Comment 5.** *Turing machines are typically defined with infinite number of tape cells, and are allowed to run arbitrarily long before halting. However, for every given input length, any computable function always uses a fixed memory and run-time (which depend on the input length).*

We now show any Turing machine with fixed memory and run-time can be simulated by an ARDT:

> **Theorem 6.** *Let $\mathbb{D} = \Sigma \cup Q \cup \{\langle \text{PAD} \rangle, \langle \text{SEP} \rangle\}$[2]. Then, $\mathcal{F}_{M,T}^{\text{Turing}}$ can be simulated by ARDTs of size $O\left(|\mathbb{D}|^4\right)$, depth $O(\log |\mathbb{D}|)$ and context length $L = M + 3$, with length complexity $O(MT)$.*

To prove the result, we show that an ARDT can compute the state of the Turing machine at each iteration. Specifically, we encode the state of the machine as a sequence of tokens from $\mathbb{D}$, where we put a token $q \in Q \subseteq \mathbb{D}$ indicating the state of the head before the token that the head reads. This way, the transition between states is a function that only depends locally on the tokens surrounding the position of the head, where all other (non-state) tokens can be copied as-is from one state to the next. Similarly to the proof in the previous section, this operation can be realized by a small *sliding-window* tree. The full proof is given in Appendix A.

### 3.4 Simulating Sparse Circuits

A circuit $\mathcal{C}$ over some alphabet $\Sigma$ is defined as a directed-acyclic-graph (DAG), with $n$ input nodes and one output node. Each internal (non-input) node with $k$ incoming edges corresponds to some function $g : \Sigma^k \to \Sigma$ computed by the node over its incoming inputs. For some input $x \in \Sigma^n$, the output of the circuit $\mathcal{C}$ is the value of the output node, when setting the input nodes of $\mathcal{C}$ to $x_1, \ldots, x_n$. The size of the circuit $\mathcal{C}$ is the number of nodes in the computational graph. We say that $\mathcal{C}$ is $k$-sparse, if the maximal in-degree of every node in the graph is $k$. Denote by $\mathcal{F}_{N,k}^{\text{Circuit}}$ the class of functions computed by $k$-sparse circuits of size $N$.

We note that sparse circuits are an extension of sparse Boolean circuits, and so can represent Turing machines with bounded memory (Arora & Barak, 2009). In this sense, this class is "equivalent" to the class of functions computed by Turing machines. However, some functions may be more efficient to compute using sparse circuits, and so it is interesting to understand how ARDTs can directly simulate sparse circuits, as demonstrated in the following theorem:

> **Theorem 7.** *Let $\mathbb{D} = \Sigma \cup \{\langle \text{PAD} \rangle\}$. Then, $\mathcal{F}_{N,k}^{\text{Circuit}}$ can be simulated by ARDTs of size $O\left(N |\mathbb{D}|^k \log |\mathbb{D}|\right)$ and context length $L \geq N$, in length complexity $O(N)$.*

---

[2] We introduce a new separator token $\langle \text{SEP} \rangle$, that is used during the generation of the ARDT, but is not part of the alphabet or state set of the Turing machine.

*Proof of Theorem 7.* Consider a $k$-sparse circuit $\mathcal{C}$ with $N$ total nodes, where $N - n$ are internal nodes. Let $g_1, \ldots, g_{N-n} : \Sigma^k \to \Sigma$ be the functions computed at the internal nodes, ordered topologically so that each function depends only on the inputs or the results of preceding nodes. Let $g_{N-n}$ denote the function computed by the output node.

Define $f_i : \Sigma^{n+i-1} \to \Sigma$ as the output of the $i$-th node in this ordering, considering all inputs and outputs from previous nodes. Each $f_i$ is effectively a $k$-Junta. By Lemma 10, there exists a decision tree $\mathcal{T}_i$ of size $O\left(|\mathbb{D}|^k\right)$ such that $\mathcal{T}_i(\Psi(\boldsymbol{x})) = f_i(\boldsymbol{x})$ for all $\boldsymbol{x} \in \Sigma^{n+i-1}$.

To accommodate inputs $\boldsymbol{x} \in \Sigma^N$, we modify each tree $\mathcal{T}_i$ to ignore the first $N - n - i + 1$ inputs. This adaptation does not affect the size of the tree.

Let $\boldsymbol{z} = \Psi(\langle \text{PAD} \rangle) \in \{0, 1\}^d$. Construct a tree as follows: begin with the rightmost branch of the tree, using functions $h_{1,1}, \ldots, h_{1,d}, \ldots, h_{N-n,1}, \ldots, h_{N-n,d}$. For each node $i \in [N - n]$ and each bit $j \in [d]$, define:

$$h_{i,j} = \begin{cases} \mathbb{1}\{\Psi(\boldsymbol{x})_{i,j} \geq 1\} & \text{if } z_j = 1, \\ \mathbb{1}\{\Psi(\boldsymbol{x})_{i,j} < 1\} & \text{if } z_j = 0. \end{cases}$$

Attach tree $\mathcal{T}_{N-n-i+1}$ at each left node $(i, j)$.

Observe that during the $i$-th iteration, the string begins with $N - n - i \langle \text{PAD} \rangle$ tokens, allowing $\mathcal{T}_i$ to process the pertinent part of the input. After $N - n$ iterations, the constructed tree calculates the output token as specified by $\mathcal{C}$. $\qquad \square$

# 4 Experiments

In this section, we experimentally validate the capabilities of ARDTs as demonstrated in the previous section and prove their language modeling potential. In Section 4.2, we first train a model based on ARDTs and test its ability to continue stories on Tinystories Eldan & Li (2023), which involves extending narratives similar to a finite state automaton. ARDTs generate coherent text that builds on existing stories, also requiring the interpretation of complex contexts and emotions. This showcases the effectiveness of sparse circuits in managing significant yet limited inputs.

Additionally, in Section 4.3, we assess the model's reasoning abilities on the Big-Bench-Hard Suzgun et al. (2022) dataset, where tasks often involve evaluating the truthfulness of propositions, effectively emulating a Turing machine as it processes inputs to determine a definitive outcome (true or false).

## 4.1 Setting

To align with the theory section, we designed our experiments to closely mirror the theoretical settings as closely as possible. We here provide a detailed description of our implementation of Auto-regressive Decision Trees (ARDTs) for next-token prediction tasks. Our objective is to utilize ARDTs as a language model that receives a sequence of input tokens $x_1, \ldots, x_n$ and predicts the subsequent token $x_{n+1}$. Initially, we employ a Word2Vec embedding Mikolov et al. (2013), denoted by $\Psi$, to convert the sequence tokens into word embeddings $\Psi(x_1), \ldots, \Psi(x_n), \Psi(x_{n+1}) \in \mathbb{R}^{100}$. We then compute a weighted average of these embeddings with exponential decay, prioritizing the most recent tokens: $\overline{\boldsymbol{v}} = \sum_{i=1}^{n} \alpha^{n-i+1} \Psi(x_i)$, where $\alpha \in (0, 1)$. Using XGBoost Chen & Guestrin (2016), we train an ensemble of decision trees, $\mathcal{T}$, which takes the input vector $\overline{\boldsymbol{v}}$ and predicts the embedding of the next token $\Psi(x_{n+1})$, aiming to minimize the mean squared error (MSE) loss. We train this model using sequences of varying lengths sampled from our dataset. During inference, the model generates text auto-regressively. At each step, it receives the current sequence $\overline{\boldsymbol{v}}$, outputs the predicted embedding of the next token $\hat{\boldsymbol{u}} = \mathcal{T}(\overline{\boldsymbol{v}})$, and identifies the token whose embedding is closest to this prediction, i.e., $\hat{x} = \arg\min_x \|\Psi(x) - \hat{\boldsymbol{u}}\|_2$. This token is then used as the next token in the sequence. The input vector is updated with the new token using $\overline{\boldsymbol{v}} \leftarrow \alpha \overline{\boldsymbol{v}} + \Psi(\hat{x})$, and the process repeats for the next iteration. Figure 2 illustrates the training and inference pipeline.

**Comment 8.** *We note that the setting described above deviates from the theory setting. 1) While the theoretical analysis focuses on the representational power of a single auto-regressive decision tree, the experiments utilize ensembles of decision trees. Notably, tree ensembles are more expressive, which suggests that our positive findings should also extend to these ensembles. 2) For simplicity, our theoretical study examines trees that generate a single output token in each iteration, rather than producing a word vector, which is the approach used in the experiments. 3) The decision trees*

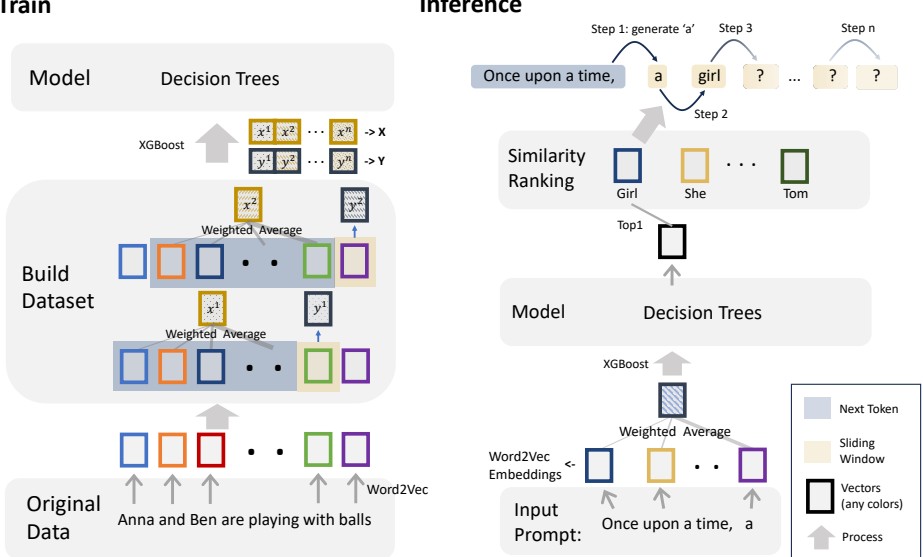

Figure 2: **The Pipeline of Our Method. (a) Training.** First, we employ a Word2Vec model to convert words into embeddings. Next, we utilize a sliding window approach to construct a dataset for training decision trees. Within this window, we performed a weighted average calculation, and the following token after the window was used as the label. **(b) Inference.** We use our trained Decision Trees for the purpose of next-token prediction.

*discussed theoretically operate on concatenated token vectors within a sliding window, in contrast to the use of vector averages in the experimental setting.*

## 4.2 The Ability to Generate Coherent Stories

We test ARDTs' ability to generate stories with the *TinyStories* Eldan & Li (2023) dataset, which is a widely-used high-quality synthetic dataset of short stories that contain words that a 3 to 4-year-old child can understand, generated by GPT-3.5 and GPT-4. Details can be found in Appendix B.2.

For experiments conducted on TinyStories, we strictly follow Eldan & Li (2023) and employ the multidimensional score provided by GPT-4, as detailed in Appendix B.5.

For baselines to compare with ARDTs, we selected several Transformer-based models. These include two small Transformers trained on the TinyStories dataset (TinyStories-1M and TinyStories-33M Eldan & Li (2023)), as well as GPT-4 OpenAI et al. (2023), to illustrate the performance differences between non-neural network methods and the Transformer architecture.

For our evaluation, we provide the models with 100 story beginnings (refer to examples in Appendix B.4), each consisting of fewer than 6 words, generated by GPT-4. We use these beginnings as inputs to the model, allowing the it to perform next token prediction, ultimately generating outputs of 20 words. For the ground truth row in Table 1, we grade complete stories from the dataset.

As shown in Table 1, ARDTs achieved performance comparable to GPT-4 and TinyStories-33M on four metrics: grammar, creativity, consistency, and plot. Our model outperforms TinyStories-1M, a Transformer-based model with 1M parameters, despite being smaller in size. These results demonstrate that although tree-based models are generally considered inferior to large neural networks, surprisingly, they can compete with small Transformers when trained on the TinyStories dataset.

## 4.3 Evaluating ARDTs in Language Reasoning Tasks

We now explore the potential of using decision trees for logical reasoning tasks using the Big-Bench-Hard dataset. The Big-Bench-Hard dataset, detailed in Appendix B.2, contains 23 challenging

Table 1: Experiment Results on TinyStories: The results show that an auto-regressive tree can achieve better performance as the GPT-Neo architecture and exhibit competitive performance compared to both GPT-4 and TinyStories-33M.

| | Model Architecture | Parameters* | Grammar† | Creativity† | Consistency† | Plot† |
|---|---|---|---|---|---|---|
| TinyStories-1M | GPT-Neo | 1M | 4.42 | 2.70 | 6.32 | 3.65 |
| TinyStories-33M | GPT-Neo | 33M | 7.80 | 6.87 | 9.10 | 7.65 |
| GPT-4 | GPT-4 | 1800B | 9.93 | 8.51 | 9.32 | 8.24 |
| Ground Truth | / | / | 8.21 | 6.32 | 7.87 | 7.56 |
| ARDTs (Ours) | Decision Tree | 0.3M | 7.85 | 4.10 | 7.36 | 5.39 |

\* For our decision trees, we report the total number of tree nodes in the ensemble as the parameter count.
† To minimize the impact of inconsistency on our results and enhance the robustness of our evaluation metrics, we calculated the average scores from ten assessments for each of the 100 stories. Each story was evaluated ten times using the same prompt provided to GPT-4.

reasoning tasks from the BIG-Bench benchmark. We selected four representative reasoning tasks for evaluation, with examples provided in Appendix B.2.

Each task involves training a separate decision tree ensemble. These ensembles utilize a weighted average of input word embeddings, as described in Section 4.1, using the word embedding layer from a pre-trained GPT-2 model trained on WebText. Each model is trained with 200 examples and tested on 50 examples. We also experiment with decision trees trained on top of a pre-trained GPT-2 Transformer model, where the output vectors from GPT-2 serve as input features for the decision trees, combining GPT-2's advanced language understanding with the analytical capabilities of decision trees.

For establishing baselines, we follow the methodology of Suzgun et al. (2022) and use accuracy as the metric. InstructGPT, Codex, and PaLM 540B are used as baselines.

As presented in Table 2, our model demonstrates substantial effectiveness in reasoning tasks, with performance comparable to state-of-the-art methods. For instance, we observe improvements of 7.4% in Boolean Expression tasks, 2% in Navigate tasks, and 7.8% in Sports Understanding tasks. Moreover, we find that further enhancements are possible by integrating decision trees with the GPT-2 Transformer, underscoring the significant impact of word embeddings on performance. However, his paper focuses on highlighting the potential of the ARDTs architecture, not word embeddings. Our results show that the ARDTs model has strong reasoning abilities.

Table 2: Experimental Results on BIG-Bench-Hard. Lin: Linear Embedding; GPT: GPT-2 Embedding. The results demonstrate that ARDTs possess good reasoning capabilities.

| BIG-Bench Hard | Srivastava et al. (2023) | | Human-Rater | InstructGPT | Codex | PaLM 540B | Ours | |
|---|---|---|---|---|---|---|---|---|
| | Random | SOTA | | | | | Lin | GPT |
| Boolean Expressions | 50 | 68.5 | 79.4 | 90 | 88.4 | 83.2 | 72.0 | 85.3 |
| Navigate | 50 | 56 | 81.9 | 68 | 50.4 | 62.4 | 55.4 | 69.2 |
| Web-of-Lies | 50 | 59.6 | 81.3 | 51.6 | 51.6 | 51.2 | 53.2 | 71.1 |
| Sports Understanding | 50 | 68.1 | 70.8 | 71.6 | 72.8 | 80.4 | 72.3 | 83.9 |
| All Tasks (avg) | 50 | 63.1 | 78.4 | 70.3 | 65.8 | 69.3 | 63.2 | 77.4 |

## 5   Discussion

The findings in this paper demonstrate that tree-based models have potential in language generation. Although they do not yet match the performance of large language models, they possess certain advantages that make them valuable for studying the emergence of intelligence on a smaller scale. Decision trees are easier to interpret (see Appendix C for more on interpretability using ARDTs), simpler to understand and analyze mathematically, and fast to train. Moreover, unlike standard neural

networks, the inference time for decision trees typically increases *logarithmically* with their size: a tree with depth $d$ can have $2^d$ nodes but only requires traversing $O(d)$ nodes per input.

This paper serves as a preliminary exploration into using ARDTs for language modeling tasks. We aim to inspire further research that integrates tree-based models into current language model pipelines, leveraging their unique strengths to enhance language generation capabilities. We believe incorporating tree-structured models into hybrid models with Transformers could be a promising direction for future research.

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

# A   Additional Proofs

For any $\mathbb{D}$, let $d = \lceil \log(|\mathbb{D}|) \rceil + 1$ and let $\Psi : \mathbb{D} \to \{0,1\}^d$ be a one-to-one mapping of tokens to Boolean vectors, s.t. $\Psi_1(s) = 1$ for all $s \in \mathbb{D}$.

**Definition 9.** *A function $f : \mathbb{D}^L \to \mathbb{D}$ is called k-Junta if there exists a set of separate indexes $i_1, \ldots, i_k \in [L]$ and function $g : \mathbb{D}^k \to \mathbb{D}$ s.t. $f(\boldsymbol{x}) = g(x_{i_1}, \ldots, x_{i_k})$.*

**Lemma 10.** *For every k-Junta $f : \mathbb{D}^L \to \mathbb{D}$, there exists a tree $\mathcal{T}$ of size $O\left(|\mathbb{D}|^k\right)$ and depth $O(k \log |\mathbb{D}|)$ s.t. $\mathcal{T}(\Psi(\boldsymbol{x})) = f(\boldsymbol{x})$ for all $\boldsymbol{x} \in \mathbb{D}^L$.*

*Proof.* Let $\mathcal{T}$ the perfect binary tree of depth $dk$, where each level of the tree corresponds to a pair $(j, l) \in [k] \times [d]$, and all the nodes at the level implement the condition $\Psi_l(x_{i_j}) \geq 1$. Observe that in this construction, each leaf correspond to a specific choice of values for $\Psi(x_{i_1}), \ldots, \Psi(x_{i_k})$, and we can set its output to be $g(x_{i_1}, \ldots, x_{i_k})$. $\qquad\square$

*Proof of Theorem 3.* Let $\mathcal{A}$ be some automaton, defined by transition function $\delta : Q \times \Sigma \to Q$, and we can arbitrarily extend it to $\delta : \mathbb{D}^2 \to \mathbb{D}$ s.t. $\delta(x, \langle \mathrm{PAD} \rangle) = q_0$ for all $x \in \mathbb{D}$. Then, from Theorem 10 there exists some tree $\mathcal{T}$ of size $O(|\mathbb{D}|^2)$ s.t. for all $\boldsymbol{x} \in \mathbb{D}^L$ it holds that $\mathcal{T}(\Psi(\boldsymbol{x})) = \delta(x_L, x_{L-n})$.

We prove by induction that for all $i \in [n]$ it holds that $\mathcal{T}_i^{\mathrm{AR}}(\boldsymbol{x}) = q_i$, where $q_i$ is the state of the automaton $\mathcal{A}$ at iteration $i$.

- Let $\boldsymbol{z} \in \mathbb{R}^{L,d}$ be the padded output of $\Psi(\boldsymbol{x})$, i.e. $\boldsymbol{z} = [\Psi(\langle \mathrm{PAD} \rangle), \ldots, \Psi(\langle \mathrm{PAD} \rangle), \Psi(x_1), \ldots, \Psi(x_n)]$. Note that since $x_{L-n} = \langle \mathrm{PAD} \rangle$ we have $\mathcal{T}_1^{\mathrm{AR}}(\boldsymbol{x}) = \mathcal{T}(\boldsymbol{z}) = \delta(x_L, \langle \mathrm{PAD} \rangle) = q_1$.

- Assume that $\mathcal{T}_{1:i-1}^{\mathrm{AR}}(\boldsymbol{x}) = (q_1, \ldots, q_{i-1})$. Therefore,

$$\mathcal{T}_i^{\mathrm{AR}}(\boldsymbol{x}) = \mathcal{T}(\Psi(\langle \mathrm{PAD} \rangle, \ldots, \langle \mathrm{PAD} \rangle, x_1, \ldots, x_n, q_1, \ldots, q_{i-1}))$$
$$= \delta(q_{i-1}, x_i) = q_i$$

Therefore, the required follows. $\qquad\square$

*Proof of Theorem 6.* We encode the state of the Turing machine by a string $\boldsymbol{s} \in \mathbb{D}^{M+1}$ as follows: if the head is in state $q \in Q$ and at position $i \in [M]$, and the memory is $m_1, \ldots, m_M \in \Sigma$, we set $\boldsymbol{s} = (m_1, \ldots, m_{i-1}, q, m_i, \ldots, m_M)$. That is, we add a token indicating the state of the head *before* the cell where the head is located. Let $\delta : Q \times \Sigma \to Q \times \Sigma \times \{\langle \mathrm{LEFT} \rangle, \langle \mathrm{RIGHT} \rangle\}$ be the transition function of the Turing machine. We define the following function $g : \mathbb{D}^4 \to \mathbb{D}^4$:

$$g(\boldsymbol{s}) = \begin{cases} x_2 & \text{if } x_1, x_2, x_3 \notin Q \\ q & \text{if } x_1 \in Q \text{ and } \delta(x_1, x_2) = (q, \alpha, \langle \mathrm{RIGHT} \rangle) \\ \alpha & \text{if } x_1 \in Q \text{ and } \delta(x_1, x_2) = (q, \alpha, \langle \mathrm{LEFT} \rangle) \\ \alpha & \text{if } x_2 \in Q \text{ and } \delta(x_2, x_3) = (q, \alpha, \langle \mathrm{RIGHT} \rangle) \\ x_1 & \text{if } x_2 \in Q \text{ and } \delta(x_2, x_3) = (q, \alpha, \langle \mathrm{LEFT} \rangle) \\ x_2 & \text{if } x_3 \in Q \text{ and } \delta(x_3, x_4) = (q, \alpha, \langle \mathrm{RIGHT} \rangle) \\ q & \text{if } x_3 \in Q \text{ and } \delta(x_3, x_4) = (q, \alpha, \langle \mathrm{LEFT} \rangle) \end{cases}$$

Observe that the function $f : \mathbb{D}^{M+1} \to \mathbb{D}^{M+1}$ s.t. $f_i(\boldsymbol{s}) = g(s_{i-1}, s_i, s_{i+1}, s_{i+2})$ exactly defines the transition between the encoded states of the Turing machine. Namely, if the state of the machine at iteration $i$ is $\boldsymbol{s}$, then the state at iteration $i + 1$ is $f(\boldsymbol{s})$. We slightly modify $g$ to handle the generation of the first iteration, as follows:

$$\tilde{g}(\boldsymbol{s}) = \begin{cases} \langle \mathrm{SEP} \rangle & x_1 = \langle \mathrm{PAD} \rangle \text{ and } x_2 = \langle \mathrm{PAD} \rangle \text{ and } x_3 = \langle \mathrm{PAD} \rangle \\ q_0 & x_1 = \langle \mathrm{PAD} \rangle \text{ and } x_2 = \langle \mathrm{PAD} \rangle \text{ and } x_3 \neq \langle \mathrm{PAD} \rangle \\ \langle \mathrm{SEP} \rangle & x_2 = \langle \mathrm{SEP} \rangle \\ g(\boldsymbol{s}) & \text{otherwise} \end{cases}$$

Now, from Lemma 10 there exists a tree $\mathcal{T}$ of size $O(|\mathbb{D}|^4)$ s.t. $\mathcal{T}(\Psi(\boldsymbol{x})) = \tilde{g}(x_1, x_2, x_3, x_4)$.

Let $\boldsymbol{s}_1, \ldots, \boldsymbol{s}_T \in \mathbb{D}^{M+1}$ the encodings of the state of the Turing machine at iterations $1, \ldots, T$. Let $\boldsymbol{x} \in \mathbb{D}^L$ be the encoding of the input, staring with $\langle \text{PAD} \rangle$ tokens, followed by one $\langle \text{BOS} \rangle$ token and the input string. Denote the output of the ARDT $\mathcal{T}^{\text{AR}}$ after $T \cdot (M+2)$ given the input $\boldsymbol{x}$, where we split the output into chunks of size $M+2$ by:

$$\mathcal{T}^{\text{AR}}(\boldsymbol{x}) = (\boldsymbol{z}_1, \ldots, \boldsymbol{z}_T) \in \mathbb{D}^{T \cdot (M+2)}, \; \boldsymbol{z}_i \in \mathbb{D}^{M+2}$$

**Claim:** For all $i \in [T]$, it holds that $\boldsymbol{z}_i = (\langle \text{SEP} \rangle, \boldsymbol{s}_i)$.

**Prove:** We prove by induction on $i$.

- For $i = 1$, notice that the input begins with 3 $\langle \text{PAD} \rangle$ tokens, followed by the input tokens $x_1, \ldots, x_M$, and therefore by definition of $\tilde{g}$ we get $\boldsymbol{z}_1 = (\langle \text{SEP} \rangle, q_0, x_1, \ldots, x_M) = (\langle \text{SEP} \rangle, \boldsymbol{s}_1)$.

- Assume the required holds for $i$. First, observe that

$$z_{i+1,1} = \mathcal{T}(\Psi(s_{i-1,M+1}, \langle \text{SEP} \rangle, s_{i,1}, \ldots, s_{i,M+1})) = \langle \text{SEP} \rangle$$

  Now, assume that $\boldsymbol{z}_{i+1,1:j} = (\langle \text{SEP} \rangle, s_{i+1,1}, \ldots, s_{i+1,j-1})$. Therefore

$$\begin{aligned} z_{i+1,j+1} &= \mathcal{T}(\Psi(s_{i,j-1}, s_{i,j}, s_{i,j+1}, \ldots, s_{i,M+1}, \langle \text{SEP} \rangle, s_{i+1,1}, \ldots, s_{i+1,j-1})) \\ &= g(s_{i,j-1}, s_{i,j}, s_{i,j+1}, s_{i,j+2}) = s_{i+1,j} \end{aligned}$$

  and by induction we get $\boldsymbol{z}_{i+1} = (\langle \text{SEP} \rangle, \boldsymbol{s}_{i+1})$

Therefore, $\mathcal{T}$ outputs the final token of iteration $T$ after $T(M+2)$ steps of auto-regression, which proves the theorem. $\square$

## B  Additional Implementation Details

### B.1  Hardware & Computational Cost

Our experiments were conducted on a single NVIDIA A100 GPU. For the Tiny Stories experiments, the training process took approximately 1 hour, and it required about 1 second to generate 20 words during the inference phase.

### B.2  Dataset Details

**Tiny Stories.** As shown in Tab. 3, the training and validation datasets of Tiny Stories contain 147,273 and 21,990 stories, respectively. We use NLTK Bird et al. (2009) as the tokenizer to obtain 420,351,665 and 4,329,963 tokens from the training dataset. In the training dataset and validation dataset, the number of words in the vocabulary is 27,455 and 11,273, respectively.

**BIG-Bench-Hard** is a dataset contains the selection of 23 difficult tasks from the BIG-Bench. These tasks are identified by their resistance to being outperformed by prior language model evaluations when compared to the average human evaluator. The BIG-Bench-Hard tasks often demand complex, multi-step reasoning, and the use of few-shot prompting without CoT, as previously utilized in BIG-Bench evaluations Srivastava et al. (2023), significantly underrepresents the true potential and performance of language models.

Four representative reasoning tasks we select for evaluate our ARDTs:

(1) *Boolean Expressions*. Example: not (True) and (True). Answer: False.

(2) *Navigate*. Example: If you follow these instructions, will you return to the starting point? Instructions: Turn left. Take 5 steps. Turn right. Answer: No.

(3) *Web-of-Lies*. Example: Delbert tells the truth. Delfina says Delbert lies. Antwan says Delfina tells the truth. Does Delfina tell the truth? Answer: No.

(4) *Sports Understanding*. Example: Is the following sentence plausible? "Elias Lindholm beat the buzzer." Answer: No.

Table 3: Basic Information about the Tinystories Dataset.

| | Training dataset | Validation dataset |
|---|---|---|
| The number of stories | 147,273 | 21,990 |
| The number of tokens | 420,351,665 | 4,329,963 |
| The word count of each story. | 54 - 5,498 | 63 - 4,254 |
| Vocabulary | 27455 | 11274 |

## B.3 Details about the Visualization of the Decision Trees

To enable visualization that treats words as features, as shown in Algorithm 1, we map word embeddings into a lower-dimensional space. This process utilizes three primary inputs: word embeddings $W$ in an $N \times 100$ matrix, where $N$ represents the number of words and 100 the dimensionality of each embedding; cluster centers $C$ in a $20 \times 100$ matrix, indicating 20 clusters within the 100-dimensional embedding space; and a mapping matrix $M$ sized $100 \times 20$, designed to reduce the embeddings' dimensionality to 20. The algorithm begins with an orthogonalization procedure, applying QR decomposition to the transpose of $C$ ($C^T$) and returning the first 20 columns of $Q^T$, thereby establishing an orthogonal basis for the cluster space. It then projects the word embeddings $W$ into this lower-dimensional space by multiplying them with the mapping matrix $M$. By iterating over each word embedding in $W$, the algorithm applies this projection and ultimately returns a set of transformed embeddings $\{E_1, \ldots, E_N\}$, where each $E_i$ provides a lower-dimensional representation of the corresponding word embedding. This approach allows us to treat each vector value as individual words, facilitating a more intuitive understanding of the data.

---

**Algorithm 1** Map Word Embeddings to Lower Dimensional Space

---

**input**
  Word Embeddings $W \subseteq \mathbb{R}^{N \times 100}$
  Cluster Centers $C \subseteq \mathbb{R}^{20 \times 100}$
  Mapping Matrix $M \subseteq \mathbb{R}^{100 \times 20}$
**procedure** ORTHOGONALIZE($C$)
  $Q, R \leftarrow$ QR Decomposition($C^T$)
  **return** $Q[:, :20].T$
**end procedure**

**procedure** MAPEMBEDDINGS($W$, $M$)
  **return** $W \cdot M$
**end procedure**

$M \leftarrow$ ORTHOGONALIZE($C$)
**for** each word embedding $w_i$ in $W$ **do**
  $E_i \leftarrow$ MAPEMBEDDINGS($w_i$, $M$)
**end for**
**Return** $\{E_1, \ldots, E_N\}$

---

## B.4 The Story Beginnings Used in Our Evaluation

In Table 6, we provide 10 examples over 100 story beginnings we used for our experiments on Tiny Stories.

## B.5 An Example on Multidimensional Scores

As an example, we took the story that appears in the example above and cut it after the word "loud". We generated a completion of stories using our ARDTs and baselines, then we prompted GPT-4 to first provide a verbal assessment and then grade the completion. The transcript of interaction is as follows, with the parts produced by GPT-4 in italic font:

Table 4: Generated Chain-of-Thought data for boolean expression.

| Input Prompt | not not ( not ( False ) ) is | True and False and not True and True is |
|---|---|---|
| Generated Prompt | Let's think step by step. Remember that (i) expressions inside brackets are always evaluated first and that (ii) the order of operations from highest priority to lowest priority is "not", "and", "or", respectively. We first simplify this expression "Z" as follows: "Z = not not ( not ( False ) ) = not not ( A )" where "A = not ( False )". Let's evaluate A: A = not ( False ) = not False = True. Plugging in A, we get: Z = not not ( A ) = not not (True) = not not False = True. So the answer is True. | Let's think step by step. Remember that (i) expressions inside brackets are always evaluated first and that (ii) the order of operations from highest priority to lowest priority is "not", "and", "or", respectively. We first simplify this expression "Z" as follows: "Z = True and False and not True and True = A and B" where "A = True and False" and "B = not True and True". Let's evaluate A: A = True and False = False. Let's evaluate B: B = not True and True = not (True and True) = not (True) = False. Plugging in A and B, we get: Z = A and B = False and False = False. So the answer is False. |

Table 5: Generated Chain-of-Thought data for navigation.

| Input Prompt | If you follow these instructions, do you return to the starting point? Always face forward. Take 2 steps right. Take 9 steps left.Take 7 steps right. | If you follow these instructions, do you return to the starting point? Turn around. Take 1 step. Take 6 steps. Turn around. Take 6 steps. Take 9 steps. Take 1 step. |
|---|---|---|
| Generated Prompt | Let's think step by step.We start at the origin (0, 0), facing the positive y-axis.(1) Always face forward: (0, 0), facing the positive y-axis.(2) Take 2 steps right: (0, 2), facing the positive y-axis.(3) Take 9 steps left: (0, -7), facing the positive y-axis.(4) Take 7 steps right: (0, 7), facing the positive y-axis. Since (0, 0) is (0, 0), we are indeed where we started. So the answer is Yes. | Let's think step by step. We start at the origin (0, 0), facing the positive y-axis. (1) Turn around: (0, 0), facing the negative y-axis. (2) Take 1 step: (0, -1), facing the negative y-axis. (3) Take 6 steps: (0, -7), facing the negative y-axis. (4) Turn around: (0, -7), facing the positive y-axis. (5) Take 6 steps: (0, -1), facing the positive y-axis. (6) Take 9 steps: (0, 8), facing the positive y-axis. (7) Take 1 step: (0, 9), facing the positive y-axis. Since (0, 9) is not (0, 0), we are not where we started. So the answer is No. |

Table 6: Examples of the story beginnings generated by GPT-4. In the inference phase, we fed these story beginnings generated by GPT-4 into the ARDTs to continue writting the stories.

| | The beginning of the stories (GPT-4 generated) |
|---|---|
| 1 | Kitty found a bright, lost treasure. |
| 2 | Thirsty, they sought a hidden pond. |
| 3 | Roxy climbed, seeking icy hill's peak. |
| 4 | Billy's idea sparked an unlikely friendship. |
| 5 | Leaves underfoot, Roxy ascended triumphantly. |
| 6 | Daisy, small yet happy, met Max. |
| 7 | Max's playful antics delighted tiny Daisy. |
| 8 | Sue's thoughtful gesture warmed her mother. |
| 9 | A candle's flame united mother, daughter. |
| 10 | The sad cow wished for companionship. |

**the following exercise, the student is given a beginning of a story. The student needs to complete it into a full story. The exercise tests the student's language abilities and creativity. The symbol ∗ ∗ ∗ marks the separator between the prescribed beginning and the student's completion**:

Once upon a time, *** there was a little girl named Lily. She loved to play outside in the sunshine. One day, she saw a big tree in the garden. She was so excited about it.

**Please provide your general assessment about the part written by the student (the one after the *** symbol). Is it grammatically correct? Is it consistent with the beginning of the story? Pay special attention to whether the student manages to complete the sentence which is split in the middle by the separator ***.**

*This is a charming and whimsical story about a little girl named Lily who feels excited upon discovering a big tree in the garden.*

*The grammar is generally correct.*

*Overall, the student's completion of the story demonstrates adequate language abilities and creativity. However, adding more details about Lily's exploration, as well as her thoughts and actions after discovering the tree, could make the story more engaging and vivid.*

***Now, grade the student's completion in terms of grammar, creativity, consistency with the story's beginning and whether the plot makes sense. Moreover, please provide your best guess of what the age of the student might be, as reflected from the completion. Choose from possible age groups: A: 3 or under. B: 4-5. C: 6-7. D: 8-9. E: 10-12. F: 13-16.***

*Grammar: 9/10, Creativity: 4/10, Consistency: 7/10, Age group: E (10-12)*

## C  Interpretability

One advantage of using decision trees over complex neural networks is the Interpretability of the computational process. Unlike neural networks, which rely on abstract vector operations in high-dimensional space, decision trees implement relatively simple logic, computing their output based on splitting criteria that may be easily displayed to, and interpreted by, humans. That said, recall that our decision trees operate on aggregations of word vector embeddings, which make interpretability harder to achieve. Specifically, each splitting rule of the decision tree is based on the value of a single coordinate, which does not necessarily have an interpretable semantic value when using rotation-invariant word embedding methods such as Word2Vec.

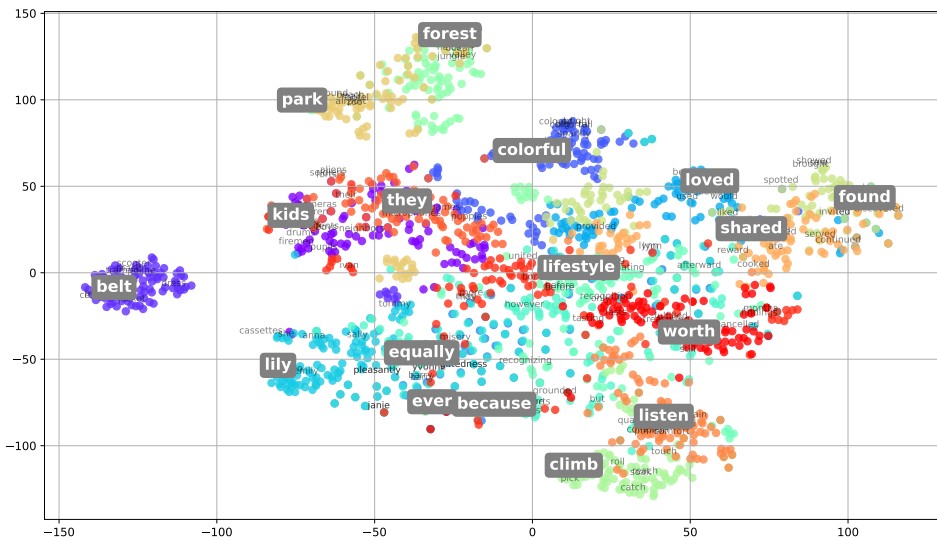

Figure 3: **t-SNE van der Maaten (2013) visualization of 20 cluster centers.** We selected 20 cluster centers and display 4 words closest to the cluster centers.

In order to generate decision trees with meaningful splitting rules, we modify the word embedding such that single coordinates have specific semantic values. To achieve this, we begin by clustering all the word vectors from the dataset (over 16K words) into 20 clusters using K-means. We then

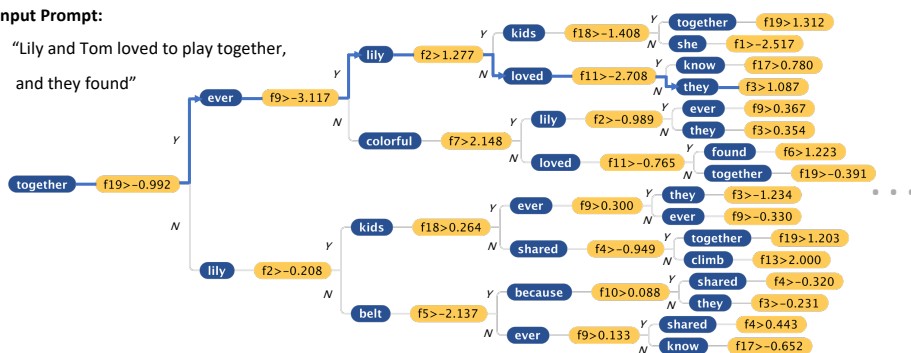

Figure 4: **Track the decision-making process within the decision trees.** We use 'Lily and Tom loved to play together, and they found' as an the input prompt and generate the next word using our ARDTs. We visualize part of the process within the decision tree. Specifically, we visualized 31 nodes of the first decision tree.

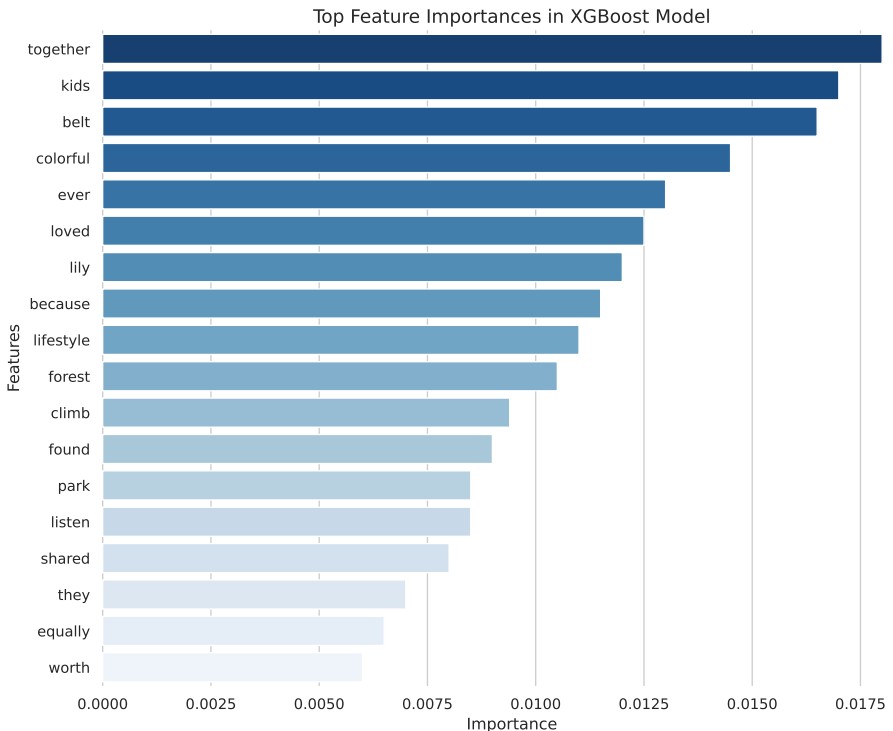

Figure 5: Feature Importance. We present the feature importance of the top 20 words most closely associated with each cluster, based on their average gain.

choose one representative word for each cluster, by taking the word that is closest to the center of the cluster in the embedding space (see Figure 3 for an illustration of the clusters and representative words). Now, these words (represented as vectors) form a basis for a *new* 20-dimensional embedding space, which is a linear subspace of the original 100-dimensional space of Word2Vec. We use these basis words to compute the new word embedding, by projecting each vector from the original space into this subspace, and representating the projection as a linear combination of the basis words. Mathematically, if $x_1, \ldots, x_k$ are the basis words, we define our new embedding $\Phi$ into $\mathbb{R}^k$ by: $\Phi(x) = \arg\min_{\mathbf{z} \in \mathbb{R}^k} \left\| \sum_i z_i \Psi(x_i) - \Psi(x) \right\|_2$. Observe that each basis word $x_i$ is mapped by $\Phi$ to a unit vector $e_i$. Intuitively, the $i$-th coordinate of the embedding $\Phi$ now represents words that are

semantically similar to the word $x_i$. Now, splitting rules based on the coordinate $i$ can be interpreted as "testing" whether a word similar to $x_i$ appears in the sentence.

We visualize one of the decision trees trained on the Tiny Stories Dataset using the new "interpretable" embedding $\Phi$ in Figure 1. Note that, unlike complex neural network architectures, which carry out opaque computations, the decision process of the ARDT with the new embedding appears to be semantically meaningful. For example, observe that the word *Lily* appears for three times as the most relevant word during node splits. Considering *Lily* is a frequently occurring name in the Tiny Stories dataset, it's frequent appearance in the tree can be deemed reasonable. We further analyze the importance of different features by plotting their importance score. We plot the importance of each cluster, represented by a single word, in Figure 5. We assess the importance of each cluster by calculating its average gain during every split within the model.

In Figure 4, we use the input sentence "Lily and Tom loved to play together and they found" as an example to visualize part of the decision-making process of the first decision tree in the ensemble. We note that each feature corresponds to a single cluster, represented by a single word, e.g. the feature $f_2$ corresponds to the word "Lily". That is, the word "Lily" will be mapped to the unit vector $e_2 = (0, 1, 0, \ldots, 0)$. Note that most words (besides the 20 words used as a basis for the embedding), will be mapped to a linear combination of the basis words, and so can also affect (positively or negatively) the value of the feature $f_2$. Since the input vector is a weighted-average of the embedding of all words, the decision when splitting on the feature $f_2$ may be affected by multiple words in the sentence.

