# OpenReview forum: "On the Power of Decision Trees in Auto-Regressive Language Modeling"
_NeurIPS.cc/2024/Conference — NeurIPS 2024 poster_

### Official Review · Reviewer_qHp8 · 2024-07-10

**Soundness:** 3
**Presentation:** 3
**Contribution:** 3
**Rating:** 5
**Confidence:** 4

**Summary:**

This paper contributes both theoretical and empirical evidence that autoregressive decision trees (ARDTs) are an expressive machine learning model. On the theoretical side, the authors show that a family of ARDTs with concatenated inputs can, in theory, efficiently (usually polynomial in size of input sequence) simulate automata, Turing machines with fixed memory and runtime, and $k$-sparse circuits. The appendix additionally proves a result of simulating $k$-Juntas. Moreover, they do provide a simple example where traditional decision trees would require exponentially more decision nodes compared to ARDTs, showing the improved expressivity of ARDTs. On the empirical side, the authors apply ARDTs on two tasks: learning to generate short stories and learning to solve a set of reasoning tasks, illustrating ARDTs can work in these settings with only limited training data and a small number of parameters.

**Strengths:**

1. While autoregressive decision trees are not new, the proven theoretical results are both novel and enlightening. I am also not aware of any such results holding for regular decision trees, increasing the potential impact of the provided results, if correct.

2. Most of the paper is well-written and easy to follow. The necessary preliminaries are explained without being too verbose, with only a couple of exceptions. In general, presentation is good.

3. Applying ARDTs to language modelling might make sense, but is non-trivial. Hence, I do acknowledge that getting ARDTs to work properly with language input might not have been easy. The results on the Big-Bench-Hard are particularly noteworthy, showing the power of neurosymbolic AI methods [1, 2].

[1] Garcez, A. D. A., & Lamb, L. C. (2023). Neurosymbolic AI: The 3 rd wave. Artificial Intelligence Review, 56(11), 12387-12406.

[2] Marra, G., Dumančić, S., Manhaeve, R., & De Raedt, L. (2024). From statistical relational to neurosymbolic artificial intelligence: A survey. Artificial Intelligence, 104062.

**Weaknesses:**

1. While the proof sketch of Theorems 3 and the full proof of Theorem 4 are very clear and well written, the remainder of the proofs are not as clear and can be hard to follow. Moreover, I have doubts about their correctness:
     + In the proof of Theorem 3 (Appendix A), I fail to see how Lemma 10 correctly applies. The assumption being that $\delta$ is a 2-Junta and a function from $\mathbb{D}^L \rightarrow \mathbb{D}$. However, how is $\delta$ a function from $\mathbb{D}^L \rightarrow \mathbb{D}$? $L$ is not equal to 2 it seems since $n$ can be larger than 2. Additionally, the function $\Psi: \mathbb{D} \rightarrow \{0, 1\}^d$ takes $\boldsymbol{x} \in \mathbb{D}^L$ as input. How is that possible? Do you apply $\Psi$ componentwise? And how does that lead to $(x_L, x_{L - n})$ in line 461?
     + In the same proof, the proof by induction starts from the base case at iteration 1 and says that
     $$
     \mathcal{T}_1^{AR}(\boldsymbol{x}) = \delta(x_L, PAD) = q_1
     $$
     but from the definition of the extended $\delta$ transitions, we have that $ \delta(x_L, PAD) = q_0$ and not $q_1$. Hence this would not prove the induction basis. I guess this can be easily solved by correctly defining $ \delta(x_L, PAD) = q_1$?
     + Proof of Theorem 6: First it is defined how a state of the TM is encoded as a string $\boldsymbol{s} \in \mathbb{D}^{M + 1}$, but then the function $g$, which takes those strings as input, is a function from $\mathbb{D}^4 \rightarrow \mathbb{D}^4$. What is supposed to be the input to $g$? Should it just be any sequence $\boldsymbol{x}$ of length 4?
     + Same proof of Theorem 6: The definition of $f$ also seems flawed. Look for example at $f_{M + 1}(\boldsymbol{s}) = g(s_{M}, s_{M + 1}, s_{M + 2}, s_{M + 3})$. The vector $\boldsymbol{s}$ only has $M + 1$ components, do you pad the non-existent ones?

2. The experiment described in Section 4.2 about generating stories seems to have some flaws:
     + It is mentioned that Table 1 contains average scores across 100 stories, yet this does not seem to be the case. How can every average be an exact integer? Is the variance of the scores over all stories equal to 0? The general lack of variability metrics such as standard deviation or quantiles should also be addressed. Moreover, if the variability is indeed 0, can you elaborate on why this can be the case? Does this mean the GPT-4 scores are not good metrics?
     + I do have severe concerns with using GPT-4 as an evaluator. It is well-known that LLMs are **not** consistent in their predictions, not only because of the inherent uncertainty (due to sampling) of the inference process, but also due to changing inputs. Did you do a control test to see if GPT-4 indeed scored the same texts consistently? Do you use sampling during inference? The fact that a previously published work used a similar evaluation protocol does not guarantee it is a good protocol.

     + Separate from the utilised evaluation protocol, I do not agree ARDTs perform on par with GPT-4 or even bigger transformers (line 306), seeing how especially their creativity and plot scores are quite a bit lower.

3. I like the experiment in Section 4.3, yet some of the choices made do not instil confidence and seem doubtful. In particular,
     + why did you only select 4 of the 23 reasoning tasks? Did you try any of the other tasks? Do the chosen tasks work particularly well with decision trees?

     + Why is SOTA lower than the provided baselines? Shouldn't the baselines be lower or equal than SOTA by definition?

     + Apart from chain-of-thought prompting of the LLM baselines, did you also provide a few examples to facilitate few-shot inference? Given that your models (GPT2 + ARDTs) are specifically trained on a small dataset, shouldn't the LLM baselines be given equal playing terms by giving them at least part of the training data as examples in their prompts? Not doing so does give an unfair advantage to your method.



The paper has promise, but there are too many uncertainties right now for me to be able to recommend any form of acceptance. However, I am open to significantly change my score if the authors can address my concerns and questions.

**Questions:**

I listed my questions together with their corresponding weaknesses above. Given the questions concern both theory and experiments, any change in score from my side does require a constructive answer or explanation on all above questions.

If any of my questions are unclear, please let me know and I will gladly try to make them clearer.

**Limitations:**

The authors do mention that the paper is only the beginning of a larger analysis of tree-shaped models added on top of language models/neural networks. Additionally, they do not claim ARDTs are the solution to anything, rather that they can be an interesting class of models that can work rather well. In that sense, I believe the authors address the limitations of their work to a sufficient degree.

---

> ### Author Rebuttal · Authors · 2024-08-07
>
> We first want to thank the reviewer for their thorough and constructive review of our paper. We appreciate your acknowledgment of the novel and enlightening theoretical analysis, clear writing, and the non-trivial application of ARDTs to language modeling.
>
> Q1: Confusions regarding the proof
>
> A1: Thank you for pointing out the clarifications required in our proofs. While we assure you that the proofs are correct, indeed, as you pointed out, they can definitely be better explained and clarified.
> - As you note, $\delta$ is not a 2-Junta but rather the function that we use to define the Junta. To be more accurate, let $f : D^L \to D$ s.t. $f(x) = \delta(x_L, X_{L-n})$. Then, $f$ is a 2-Junta, and from Lemma 10 there exists a tree satisfying $T(\Psi(x)) = f(x)$. Indeed, $\Psi$ is applied componentwise.
> - There is some abuse of notations in our definitions, where the state $q_i$ defines the state of the Automaton at iteration $i$, while we also use $q_0$ to define the initial state of the Automaton, so in fact it is always the case that the first state is $q_0$, i.e. $q_1 = q_0$. We realize that these notations are confusing, and will fix this by using a different notation to indicate the initial state.
> - You are correct, there is a typo and the input to $g$ should be $x$ (of length 4), and not $s$. Thanks for the catch.
> - Yes, values that “overflow” can be considered as <PAD> tokens. These do not affect the output of the function. We will clarify this in the paper.
>
> Q2.1 Why is every average of the 100 different stories an exact integer? Is the variance of the scores over all stories equal to 0?
>
> A2.1: The variance of the score is not zero. Following the Tiny-stories [1] paper, we rounded the results to integers. In the revision, we adjusted the results to two decimal places. The results are shown in the **Rebuttal Table 4**.
>
> Q2.2: Did you do a control test to see if GPT-4 indeed scored the same texts consistently?
>
> A2.2: We found that GPT-4's evaluation scores are not entirely consistent; however, the variance (less than 0.5) is considered acceptable (see **Rebuttal Table 5**). Actually, to enhance the consistency of GPT-4's scores, we followed [1] to setting up more precise and informative prompts (see **Lines 534-554 in Appendix**).
>
> As shown in **Rebuttal Table 4**, we have updated the scores in **Table 1** by averaging the results of ten evaluations for each of the 100 stories. Each story was scored using the same prompt provided to GPT-4 ten times. Blue means higher than TinyStories-1M (transformer-based model)
>
> Q2.3:  Do you use sampling during inference?
>
> A2.3:  No.
>
> Q2.4:  I do not agree ARDTs perform on par with GPT-4 or even bigger transformers (line 306)
>
> A2.4: Good point, we will make line 306 more accurate: ARDTs (\~0.3M parameters) trained on the TinyStories dataset are on par with GPT-4 (\~1800B) or even bigger transformers trained on huge internet dataset regarding the creativity and plot, but remain inferior in terms of grammar and consistency.
>
> Q3.1: Why did you only select 4 of the 23 reasoning tasks? Did you try any of the other tasks? Do the chosen tasks work particularly well with decision trees?
>
> A3.1: We just chose 4 representative tasks. The results of all 23 reasoning tasks are shown in **Rebuttal Table 6**. The results demonstrate that ARDTs possess good reasoning capabilities.
>
> Q3.2: Why is SOTA lower than the provided baselines? Shouldn't the baselines be lower or equal than SOTA by definition?
>
> A3.2: We apologize for any confusion caused. The term ‘SOTA’ here refers to the performance benchmarks borrowed from **Table 3 in the BIG-Bench-Hard [2] paper**, which represent the state-of-the-art performance in the BIG-Bench paper [3]. We will change 'SOTA' to 'SOTA Methods in BIG-Bench' in the revision.
>
> Q3.3: Apart from chain-of-thought prompting of the LLM baselines, did you also provide a few examples to facilitate few-shot inference? Given that your models (GPT2 + ARDTs) are specifically trained on a small dataset, shouldn't the LLM baselines be given equal playing terms by giving them at least part of the training data as examples in their prompts? Not doing so does give an unfair advantage to your method.
>
> A3.3: We will run this experiment with a few-shot examples and report the results in the final version.
>
> We hope that this clarifies any misunderstandings and we encourage the reviewer to increase their score if we have resolved their concerns or to let us know otherwise so we may try to clear up any remaining confusion.
>
> [1] Eldan, Ronen, and Yuanzhi Li. "Tinystories: How small can language models be and still speak coherent english?." arXiv preprint arXiv:2305.07759 (2023).
>
> [2] Suzgun, Mirac, et al. "Challenging big-bench tasks and whether chain-of-thought can solve them." arXiv preprint arXiv:2210.09261 (2022).
>
> [3] Srivastava, Aarohi, et al. "Beyond the imitation game: Quantifying and extrapolating the capabilities of language models." arXiv preprint arXiv:2206.04615 (2022).

---

> > ### Comment · Reviewer_qHp8 · 2024-08-11
> > **Acknowledgement of Author Rebuttal**
> >
> > Thank you for taking the time to clarify my concerns. In particular, my concerns about the validity of the theoretical results are now addressed, though I do hope the authors take the time to improve the writing of the proofs. I also appreciate the proposed revisions and more detailed insights when using GPT-4 as an evaluator. Please do add these insights to the main paper where possible or refer to them in the appendix of the paper as they validate your evaluation protocol. Additionally, providing the results for all 23 reasoning tasks from the BIG-Bench-Hard benchmark is greatly appreciated as well, and further eliminates some of my empirical concerns. Please do add these more complete results to the appendix of the paper as they again only strengthen your point.
> >
> > I will increase my score to "5: borderline accept" for now. If the authors can provide the numbers for the BIG-Bench-HARD reasoning tasks where GPT is given a few examples and the numbers are still somewhat favourable for ARDTs, I am willing to further increase to "6: weak accept". Apart from that, I have one other question that could, together with the results of the few-shot experiment, allow me to increase my score to a full accept.
> >
> > Why did you limit the number of "parameters" to 0.3M? Did you run into computational bottlenecks when using more than 10 trees (**Rebuttal Table 3**)? If so, this would be an additional limitation worth mentioning, as scaling ARDTs could to be an important component of even better future results.

---

> > > ### Author Response · Authors · 2024-08-13
> > > **Response to comments by Reviewer qHp8**
> > >
> > > We would like to thank you once again for your very helpful and constructive review comments.
> > >
> > > A1: Few-shot GPT-4 performance: we present the results of few-shot inference. Due to time constraints, we were unable to complete experiments for all 23 reasoning tasks, so we arbitrarily selected 2 tasks instead. As shown in the table, few-shot inference offers significantly less benefit for reasoning tasks compared to CoT, while our ARDTS has demonstrated clear advantages.
> > >
> > > |              |  | GPT-XL |         |     Ours     | |
> > > |-------------|:---:|:---:|:---:|:---:|:---:|
> > > |             | 1 shot | 2 shots | 4 shots | Lin  | GPT  |
> > > | Navigation  | 47.1   | 48.6    | 50.5    | 55.4 | 69.2 |
> > > | Web-of-lies | 50.0     | 52.3    | 51.8    | 53.2 | 71.1 |
> > >
> > > A2: Limiting the parameter count: we want to emphasize that the goal of this paper is to perform an initial study demonstrating the capabilities of decision trees for certain language modeling tasks, as they remained largely unexplored in this context until now. While we did not specifically limit the parameter count of the decision trees, and do not see computational barriers in achieving a modest increase in the number of trees, we leave to future work a thorough exploration of how decision trees can scale to match transformers that are several orders of magnitude larger. Such study may require scaling the depth, ensemble size and input and output dimension of the trees, and may be much more hardware intensive, requiring additional engineering and optimization novelties that are beyond the scope of this paper.
> > >
> > > We hope this addresses any concerns the reviewer may have. We encourage the reviewer to increase their score if their concerns have been resolved, or to let us know if there are still issues we need to clarify.

---

### Official Review · Reviewer_97Y1 · 2024-07-11

**Soundness:** 3
**Presentation:** 3
**Contribution:** 3
**Rating:** 6
**Confidence:** 2

**Summary:**

This paper explores the idea to use AutoRegressive Decision Trees for language modelling. From a theoretical perspective, ARDTs are shown to model systems such as finite automata and more generally Turing machines. From an experimental point of view, ARDTs are shown to be able to generate grammatically correct sentences, with performance similar to small transformers.

**Strengths:**

+ Novelty in the idea of using ARDTs for language modelling
+ Nice compromise between performance and interpretability
+ Interesting theoretical analysis of modeling different systems with ARDTs

**Weaknesses:**

- Experimental evaluation not aligned with the theoretical framework, as it uses tree ensembles
- Interpretability is reduced when switching from single decision trees to ensembles

**Questions:**

* The difference between the theoretical framework and the experimental setting described in Comment 8, page 7, is due to the complexity of implementing the theoretical solution, or to other issues?

**Limitations:**

* While the paper introduces ARDTs by stressing the point that they are an interpretable model, then in the experimental evaluation suddenly only tree ensembles are used. Is it because performance are otherwise much lower? Tree ensemble clearly do not have the same degree of interpretability of single decision trees.

---

> ### Author Rebuttal · Authors · 2024-08-07
>
> We first want to thank the reviewer for their thorough review and positive assessment of our paper. Specifically, we appreciate your acknowledgment of the paper's novel ideas, good balance between performance and interpretability, and interesting theoretical analysis.
>
> Q1: The difference between the theoretical framework and the experimental setting described in Comment 8, page 7, is due to the complexity of implementing the theoretical solution, or to other issues?
>
> A1: The setting we study in the theory section is not efficient to implement in practice, as it requires maintaining a decision tree with a large input size (the sliding window) and potentially generating many output tokens. As with many theoretical works, we study a simplified setting that is simpler to analyze mathematically, and while this setting is not efficient to implement in practice, it still captures the key properties of the experimental setting.
>
>
> Q2: While the paper introduces ARDTs by stressing the point that they are an interpretable model, then in the experimental evaluation suddenly only tree ensembles are used. Is it because performance is otherwise much lower? Tree ensembles clearly do not have the same degree of interpretability of single decision trees.
>
> A2: Although the focus of our paper is to demonstrate the computational capabilities of ARDTs through theoretical and empirical evidence, interpretability (as detailed in **Appendix C** in our paper) is not our main emphasis. As shown in **Rebuttal Table 3**, the performance of a single tree is indeed lower than that of tree ensembles. However, it can still be utilized for interpretability analysis.
>
>
> We hope that this clarifies any misunderstandings and we encourage the reviewer to increase their score if we have resolved their concerns or to let us know otherwise so we may try to clear up any remaining confusion.

---

> > ### Comment · Reviewer_97Y1 · 2024-08-12
> > **Rebuttal**
> >
> > I thank the authors for the replies and for the additional results presented in the rebuttal. I have raised my score to Weak Accept.

---

### Official Review · Reviewer_o5To · 2024-07-15

**Soundness:** 3
**Presentation:** 3
**Contribution:** 3
**Rating:** 7
**Confidence:** 3

**Summary:**

* This work studies theoretical and practical applications of auto-regressive decision trees (ARDT) in language generation and reasoning tasks.
* Through theoretical analysis, the authors show that ARDTs can learn more sophisticated functions than previously known, such as automata, Turing machines, and sparse circuits.
* Using ARDT with a transformer (GPT-2), they achieved performance comparable to SoTA models (InstructGPT, Codex, and PaLM 540B).

**Strengths:**

* The problem definition, theoretical analysis, model description, and experimental setup are presented clearly, making the paper accessible and informative.
* With the interpretability of decision trees, we can better understand the language generation process.

**Weaknesses:**

1. No mention of the limitations of ARDT for language modeling.
2. Performance changes with the text length. What will the performance be in generating long texts?

**Questions:**

1. How does the depth of trees affect performance and inference speed?

---

> ### Author Rebuttal · Authors · 2024-08-07
>
> Thank you for your thorough review and very positive assessment of our paper. Specifically, we appreciate your acknowledgment of the paper's clarity, informativeness, and its contribution to model interpretability.
>
> Q1: Performance changes with the text length. What will the performance be in generating long texts?
>
> A1: In our paper, we reported the performance of extending a story by 20 words. We have expanded this to 50 words and compared the trend in performance changes, as shown in the **Rebuttal Table 1**.
> Although Auto-Regressive Decision Trees (ARDTs) were not specifically designed to handle lengthy texts—a task left for future work—their performance remains robust even when expanded to 50 words. Both the transformer-based model and our method exhibit a decreasing trend as the word count increases, with ARDTs experiencing a slightly greater decline (see **Rebuttal Table 1**).
>
> Q2: How does the depth of trees affect performance and inference speed?
>
> A2: In **Rebuttal Table 2**, we show the impact of tree depth on performance and inference speed. Inference speed is measured during testing by recording the start and end times of the prediction using the time.time()function and then calculating the difference to determine the inference time. As **Rebuttal Table 2** shows, the depth of the tree is positively correlated with performance and inference speed.
>
> We hope that this clarifies any misunderstandings and we encourage the reviewer to increase their score if we have resolved their concerns or to let us know otherwise so we may try to clear up any remaining confusion.

---

> > ### Comment · Reviewer_o5To · 2024-08-13
> >
> > Thanks, Authors, for the detailed response to my queries. My concerns are addressed.

---

### Author Rebuttal · Authors · 2024-08-07

We thank all reviewers for their comprehensive and constructive feedback on our submission. We are pleased that our work is recognized for its novelty (Reviewers o5To, 97Y1, and qHp8), its novel and enlightening theoretical analysis (Reviewers o5To and 97Y1), and a nice balance between performance and interpretability (Reviewers o5To and 97Y1).

We have prepared an extensive, point-by-point response for each reviewer, outlining our plans to address their concerns and suggestions for additional analyses to enhance the manuscript. **All new tables can be found in the attached one-page PDF file**. We believe that our response will address the reviewers' concerns and allow us to promptly resolve any remaining minor issues.

Once again, we would like to thank all reviewers for their comments and we are looking forward to the discussion period.

---

### Decision · Program_Chairs · 2024-09-25

**Decision:**

Accept (poster)

**Comment:**

This paper is a solid mix of interesting theoretical and empirical results.  The paper stands in contrast to most DNN/transformer based language modeling, and as such, broadens the conversation about the space of useful models.  The paper is well-written, and the authors have improved the presentation based on a good dialogue with the reviewers. I believe this paper will be of interest to a wide variety of researchers and industry practitioners.